depression; anxiety; mental health; psychotherapy; disease outbreaks

**Corresponding author:**
Biksegn Yirdaw;
Email: Biksegn.yirdaw@lshtm.ac.uk

# Effectiveness of psychological crisis interventions during infectious disease outbreaks in low- and middle-income countries: a systematic review of Randomized Control Trials

Biksegn Asrat Yirdaw[1,2] ⬤, Jun Angelo Sunglao[3], Muhammad Alkasaby[1,2] ⬤ and Julian Eaton[1,2,4]

[1]UK Public Health Rapid Support Team, UK Health Security Agency/London School of Hygiene & Tropical Medicine, London, UK; [2]Centre for Global Mental Health, London School of Hygiene & Tropical Medicine, London, UK; [3]Mind+ Philippines and Psychological Association of Philippines, Manila, Philippines and [4]CBM Global Disability Inclusion, Cambridge, UK

## Abstract

The huge mental health treatment gap in low- and middle-income countries (LMICs) is further exacerbated when infectious disease outbreaks occur. To address the increasing mental health needs during outbreaks, the availability of flexible and efficient mental health interventions is paramount, especially in low-resource settings where outbreaks are more common. Psychological interventions may help to address these mental health needs with efficient implementation costs. However, there is a huge paucity of quality evidence to inform psychosocial interventions during outbreaks. This systematic review sought to update the existing evidence to inform the effectiveness of psychological interventions that addresses mental health issues during outbreaks in LMICs.

Six electronic databases were searched – Scopus, PubMed, PsycINFO, Embase, Cochrane library and CINAHL. We included randomised controlled trials of psychological interventions aimed to address common mental health conditions among adults affected by infectious disease outbreaks in LMICs. Studies were excluded if they were done among all age groups, used mixed interventions with pharmacotherapies, addressed severe mental health conditions and were published other than in English. The quality of evidence in the included trials was assessed using the Cochrane Collaboration risk of bias tool.

We included 17 trials that examined the effectiveness of psychological interventions among outbreak-affected adults in LMICs. The quality of studies was generally average but tended to provide evidence that brief psychoeducational interventions based on cognitive restructuring, mindfulness, relaxation and stress management techniques were effective in reducing perceived stress and anxiety symptoms, and in improving resilience and self-efficacy. Similarly, mindfulness-based interventions and mindfulness stress reduction treatments were effective in addressing depression, anxiety and generalised anxiety disorder.

Brief psychological interventions that can be delivered by non-specialists could have value in addressing the huge mental health needs in outbreak contexts.

## Impact statement

Given the increasing mental health needs during infectious disease outbreaks, psychological interventions that are culturally acceptable and effective should be given priority for under-resourced countries. This systematic review provides important insights into the benefits of psychological interventions in addressing common mental health issues of outbreak-affected populations in low- and middle-income countries (LMICs). Brief psychoeducational interventions with stress management principles were helpful in addressing common mental health issues, including depression, anxiety, post-traumatic stress symptoms and sleep problems associated with emergencies of infectious disease outbreaks in LMICs. Overall, non-specialist delivered, brief sessions are more likely to be valuable in addressing mental health issues that arise in outbreaks. Filling the existing knowledge gap with quality evidence will contribute to the development of the standardised, evidence-based and contextually relevant intervention guidelines that are applicable to LMICs. Therefore, future efforts should focus on improving access to quality data that can inform evidence-based decisions.

## Introduction

Infectious disease outbreaks have had a devastating impact on lives and livelihoods around the globe (Baker et al., 2021), and are a threat to planetary health and development (Huremović, 2019). The profound impacts of intermittent disease outbreaks include increased mortality, reduced effectiveness of health systems, social inequity and economic crisis (Sampath et al., 2021). Disease outbreaks pose a significantly increased risk to the mental health of affected individuals and communities, particularly in low- and middle-income countries (LMICs) where health system resilience is low and the treatment gap is high (Jacob, 2017).

Although the impact of infectious disease outbreaks is on a wider population, specific groups of people are particularly vulnerable, including people directly affected by the disease, people with pre-existing health conditions and disabilities and frontline healthcare workers (Singu et al., 2020). Evidence shows that the prevalence of several mental health problems such as post-traumatic stress disorder (PTSD), depression and anxiety symptoms doubled during infectious disease outbreaks and pandemics (Schindell et al., 2024; Hossain et al., 2020; Yuan et al., 2022). For instance, a 76% prevalence of PTSD symptoms and 48% prevalence of anxiety-depression symptoms were recorded during the Ebola epidemic in Siera Leone in 2015 (Jalloh et al., 2018). Similarly, a 64% prevalence of psychological distress and 40.7% prevalence of PTSD was reported among Severe Acute Respiratory Syndrome (SARS) survivors in Hong Kong in 2004 (Lee et al., 2007). The COVID-19 pandemic had a huge impact on population mental health and contributed to a more than 25% increase in cases of depression and anxiety globally (World Health Organization, 2022).

People with pre-existing mental health conditions were impacted to a greater extent than others (Boden et al., 2021). This may be for two reasons: in addition to being susceptible to the experience of stress common to everyone, mental health services are often disrupted, as occurred worldwide during the COVID-19 pandemic. Access to basic counselling services, medication adherence programmes, social support mechanisms and emergency mental health services also collapsed. The impact was more severe when countries closed schools and workspaces and imposed restrictions in movement and quarantine measures. In addition, mental health services were often de-prioritised, community services were suspended and facilities were changed to quarantine facilities (Yirdaw et al., 2024). With all the added risks to people with mental conditions, maintenance of mental health services was important, as a part of wider response measures. However, the capacity of health systems in LMICs to quickly develop plans and to respond to mental health needs was very limited and the process often is slow (Kola et al., 2021). While in some countries, online options using telemedicine or digital technology enabled mental health services to bridge some gaps, LMICs struggled to adapt and maintain mental health service delivery (Arenliu et al., 2020). For instance, during the COVID-19 pandemic in China, several key challenges were noted (Duan and Zhu, 2020): (i) little attention was given to the practical implementation of psychological interventions, (ii) little effort was made to align interventions into community healthcare services, (iii) there was a shortage of professionals and resources and (iv) there were restrictions to entry to isolation centres to receive appropriate care. During the COVID-19 pandemic in Africa, mental health interventions were not often included in planning, due to the lack of political commitment, low prioritisation of mental health during emergencies compared with other response activities and the scarcity of financial and human resources allocated to mental health activities (Yirdaw et al., 2024; Walker et al., 2022).

Implementing the established good practice of enabling frontline workers to deliver basic psychological interventions as part of other response activities was also challenging due to complicated work procedures, heavy workloads and the lack of standardised training resources (Duan and Zhu, 2020). Given the significant mental health impact of outbreaks and associated public health counter-measures, the application of evidence-based interventions with alternative treatment and support solutions should be part of outbreak response plans.

While acknowledging the contribution of previous studies (Pollock et al., 2020; Zace et al., 2021; Yang et al., 2021), there is a huge paucity of quality evidence to inform effective psychosocial interventions to address mental health issues during infectious disease outbreaks. The most recent systematic review (in 2021) of all intervention types with different study designs found a huge evidence gap where no randomised controlled trials (RCT) were carried out in LMICs (Zace et al., 2021). The lack of evidence is partially due to difficulties in implementing research in outbreak contexts, challenges in the measurement of treatment outcomes and lack of quality data on a longer impact of trials. Our systematic review explores the literature to update the existing evidence gap with a body of evidence to inform effective psychological interventions to address mental health issues during infectious disease outbreaks in LMICs.

## Methods

We searched for RCTs evaluating the effectiveness of psychosocial interventions in infectious disease outbreaks in LMICs. This systematic review is reported using the Preferred Reporting Items for Systematic Reviews and Meta-Analyses (PRISMA) guidelines.

### *Inclusion and exclusion criteria*

Psychosocial interventions are defined as strategies, activities, techniques and toolkits that address psychological and social problems and promote mental wellbeing. We used a broad definition of therapeutic practices, including but not limited to cognitive behavioural therapy, supportive therapy, interpersonal psychotherapy, counselling and mindfulness. Psychological interventions could be delivered through various means such as face-to-face modalities (whether group or one-to-one), or through the use of technology like telemedicine/teletherapy, or software-based interventions such as mobile applications.

The general inclusion criteria for this systematic review were: (i) trials with any type of psychological interventions, (ii) conducted in LMICs, (iii) studies must be RCTs, (iv) conducted among adults with age ≥18 years and (iv) carried out to address mental health conditions in infectious disease outbreaks.

Studies were excluded if they were: (i) included all age groups and not reporting on adults separately, (ii) focused on non-outbreak settings, (iii) used mixed interventions including pharmacological therapies concurrently, with no separate analysis of psychological interventions only, (iv) addressed only severe mental health conditions including psychosis and (v) published in other languages than English.

## Literature search strategies

We searched six databases (Scopus, PubMed, PsycINFO, Embase, Cochrane library and CINAHL) and other sources including the manual search of Google Scholar. There were no restrictions on publication date, study type and design in the initial search. Databases were searched in 15 to 25 October 2023 without language restrictions. The keywords used for searching were psychological interventions, mental conditions, infectious disease outbreaks and the list of LMICS. Similar concepts, synonyms and medical subject headings (MeSH) were used for each keyword. Appropriate syntax was developed and used for each database. The search strategies used for the search are available in Supplementary Appendix 1.

## Study selection

Studies identified from the search were screened by topic and exported to EndNote 20 software. Duplicates were removed from the EndNote and the remaining articles were then moved to Rayyan software for further duplicate identification and abstract screening. Studies that fulfilled most of the inclusion criteria were identified from the abstract screening. Full-text articles were searched by BAY and JAS. BAY and JAS double-checked the screened articles, resolved disagreements and assessed the full-text articles against the inclusion criteria independently.

## Data extraction and management

Data extraction was done by BAY and JAS using the Cochrane Collaboration data collection form for RCTs. The extracted data includes publication year, study setting, population, country, sample size, type of intervention, number of sessions, session duration, method of delivery, outcomes, outcome measures, key findings and limitations.

## Risk of bias assessment

Two of the authors (BAY and JAS) evaluated each study using the Cochrane Collaboration risk of bias tool (Higgins et al., 2011). The tool formalises the judgment of specific features of a randomized control trial to assist review authors in identifying possible limitations and considerations for the assessing strength of the results of an article. This tool has five key domains for assessment: selection bias, reporting bias, performance bias, detection bias and attrition bias. Each study in the risk of bias assessment was judged under each category of bias as either low risk for bias, high risk for bias or unclear. Unclear suggests a lack of sufficient information or persistent uncertainty over the potential for bias under this category.

## Data synthesis

The extracted and collated data were summarised in tables, with data captured including study design, participants, settings, sample size, intervention type, duration of each intervention and outcome measures. A narrative synthesis was done to analyse the differences, patterns and similarities of interventions. No meta-analysis was conducted due to the high heterogeneity of the trials in several aspects such as differences in the quality of the data, outcome measure, intervention type, session duration and delivering agents.

## Results

### Characteristics of the included studies

Of 10,890 screened articles, 2,809 duplicates were removed. After removal of duplicates, 5,955 articles were excluded because they did not fulfil at least one of the inclusion criteria – not outcome of interest, population of interest, intervention of interest or not a systematic review. The full text of 166 articles was reviewed to check whether they fulfilled all the inclusion criteria. In the first round of full-text review, we excluded 104 articles because they were not mental health related (76 articles), not the right population (16 articles), not an intervention (two articles) and not in English (eight articles). Finally, we selected 17 articles that fulfilled all the inclusion criteria (Figure 1).

All the included trials were conducted in five countries during the COVID-19 pandemic from 2020 to 2023: these were seven from China (Fan et al., 2021; Li et al., 2023; Li et al., 2020; Liu et al., 2021; Sun et al., 2022), six from Iran (Ghazanfarpour et al., 2022; Khosravi et al., 2022; Mirhosseini et al., 2022; Shabahang et al., 2021; Shaygan et al., 2021; Shaygan et al., 2023), two from Turkey (Dincer and Inangil, 2021; Hosseinzadeh, 2022), one from India (Gupta et al., 2021) and one from Jordan (Alkhawaldeh, 2023). As shown in Table 1, half of these trials (n = 8) were conducted among COVID-19 patients and six trials (n = 6) were among frontline healthcare workers involved COVID-19 response. The remaining studies focused on college students (n = 2) and pregnant women (n = 1). In terms of setting, 13 trials were conducted in hospital-based settings, four (n = 4) were in community-based health centres and one (n = 1) quarantine facility. The total number of study participants included in all trials was 1,687 and the sample size in each study ranged from 35 to 118. Table 1 provides an overview of the characteristics of the included studies.

### Outcome measures

Of the 17 included trials, 14 of them targeted anxiety symptoms only and 10 of them assessed both anxiety and depression as a primary outcome (Table 2). Stress, post-traumatic stress symptoms, psychological distress, resilience, burnout, sleep quality and self-efficacy were primary outcomes in one or more trials. The tools used to measure these outcomes vary significantly in type, item, validation and cut-off point. Four trials used combined tools to assess depression, anxiety and stress altogether; these were: the Depression, Anxiety and Stress Scale (DASS-21) (Gupta et al., 2021; Hosseinzadeh, 2022; Li et al., 2020) and the Hospital Anxiety and Depression Scale (HADS) (Ghazanfarpour et al., 2022). Another five trials evaluated depression independently using the Patient Health Questionnaire (PHQ-9), (Zhou et al., 2022; Sun et al., 2022) the Hamilton Depression Rating Scale (HAMD), (Liu et al., 2021) and the Self-rating Depression Scale (SDS) (Li et al., 2023; Fan et al., 2021). Several tools were used to assess anxiety independently including the State Anxiety Scale (Dincer and Inangil, 2021) Self-rating Anxiety Scale (Fan et al., 2021; Li et al., 2023), Hamilton Anxiety Rating Scale (Liu et al., 2021), COVID-19 Anxiety Questionnaire (Shabahang et al., 2021), Short Anxiety Inventory (Shabahang et al., 2021), State Trait Anxiety Inventory (Shaygan et al., 2023) and the Generalised Anxiety Disorder Questionnaire (Sun et al., 2022; Zhou et al., 2022). The lack of consistency in the use of outcome measures and a lack of clarity on the degree of cultural validation of the tools across studies was observed.

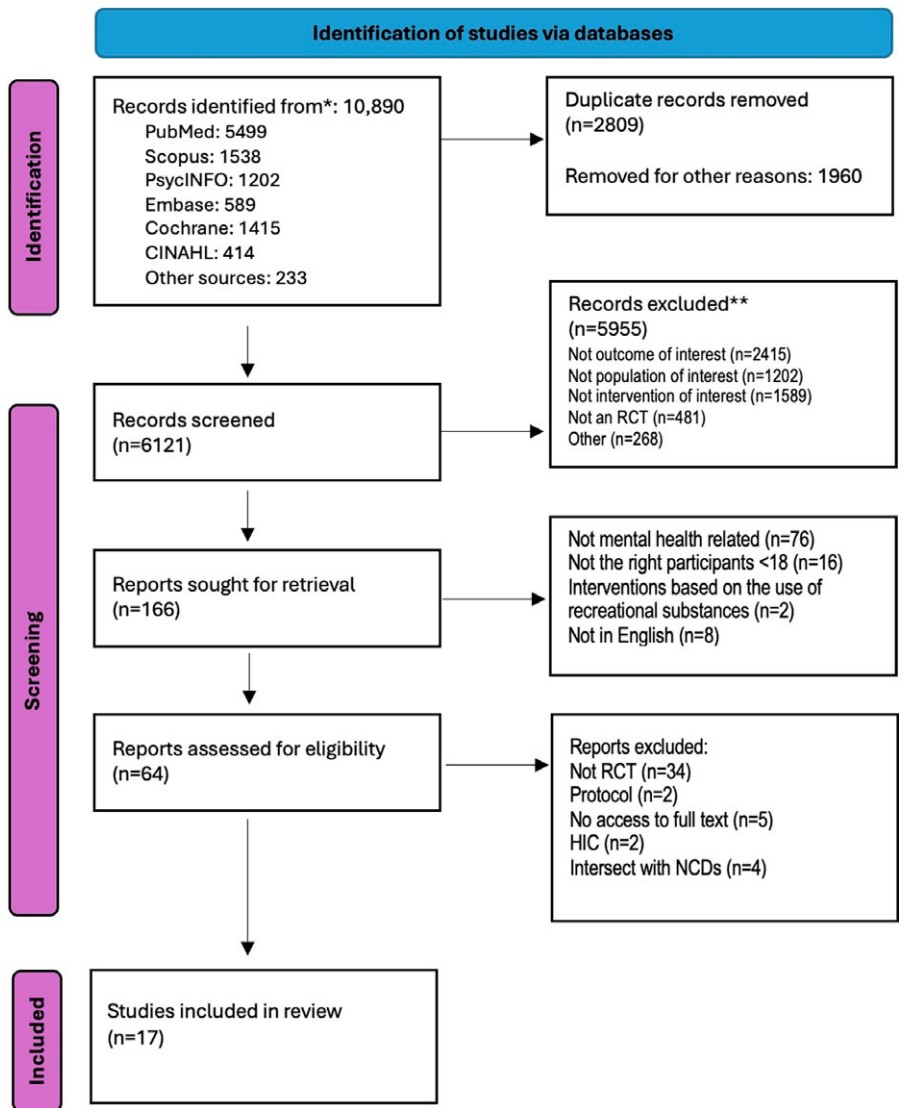

**Figure 1.** PRISMA flow diagram of search results.

### Intervention characteristics and effectiveness

Of 17 included trials, seven (n = 7) of them used CBT principles (Sun et al., 2022; Hosseinzadeh, 2022; Li et al., 2023; Ghazanfarpour et al., 2022; Shabahang et al., 2021; Liu et al., 2021), of which two (n = 2) of them combined mindfulness with CBT (Ghazanfarpour et al., 2022; Hosseinzadeh, 2022). Five (n = 5) of the included trials used psychoeducational interventions based on training, cognitive restructuring, stress management, positive therapy and relaxation techniques (Shaygan et al., 2021; Shaygan et al., 2023; Mirhosseini et al., 2022; Liu et al., 2022; Alkhawaldeh, 2023). Two more trials used mindfulness techniques alone involving practical stress reduction exercises (Sun et al., 2022; Li et al., 2023). The remaining trials used Narrative Exposure Therapy (NET) (Fan et al., 2021), Emotional Freedom Techniques (Dincer and Inangil, 2021), Brief Eclectic Psychotherapy (Gupta et al., 2021) and individual counselling (Khosravi et al., 2022).

Of the 17 included trials, 13 of them delivered interventions remotely, two were delivered face-to-face (Alkhawaldeh, 2023; Li et al., 2020) and other two used a hybrid approach (remotely and face-to-face) (Fan et al., 2021; Khosravi et al., 2022). Different

digital tools were used to deliver interventions remotely such as mobile apps, websites, telephone calls and messaging platforms like WeChat, zoom and WhatsApp. Most of these used live video calls as a means of delivering established intervention models. The interventions varied by the number and duration of sessions. Overall, the number of sessions ranged from a single to 14 sessions, lasting for 15 minutes up to 2 hours per session. The delivering agents were trained healthcare workers including psychiatrists, nurses, psychologists and mental health experts.

The most structured and intensive intervention was NET which involved up to two sessions per week with a session duration of 90–120 minutes and lasted for eight weeks. Sessions were delivered using a hybrid approach remotely (via the internet online, mobile phones, WeChat) and face-to-face in a one-to-one model in clinics (Fan et al., 2021). Study participants were followed up for 6 months after the intervention. The NET intervention was used to treat post-traumatic stress, depression and anxiety symptoms of COVID-19 patients admitted to hospitals. The intervention included three phases: (i) diagnostic interviews and psychoeducation, (ii) constructing a lifeline with a life events timeline and (iii) narrative of the exposure.

**Table 1.** Characteristics of the included studies in this systematic review (N=17)

| Author, year – country | Study population | Settings | N (intervention vs control) | Mean age (yrs) | Interventions | Controls |
|---|---|---|---|---|---|---|
| Alkhawaldeh, 2023 – Jordan | Nurses responding to COVID–19 | Community-based health centres | 84 (42:42) | 29.9 | Psychoeducational intervention that involves cognitive restructuring, relaxation and stress management techniques | Waitlist |
| Dincer and Inangil, 2021 – Turkey | Nurses caring for COVID–19 patients | University Hospital | 80 (35:45) | 33.45 | Emotional freedom techniques | Waitlist |
| Fan et al., 2021 – China | COVID–19 patients | Three COVID–19 designated hospitals | 111 (56:55) | 46.4 | Narrative exposure therapy (NET) and personalized psychological intervention | Personalized psychological treatment |
| Ghazanfarpour et al., 2022 – Iran | Healthcare providers caring for COVID–19 patients | Community-based COVID–19 clinics | 111 (55:56) | Not reported. | Cognitive-behavioural and mindfulness-based techniques | Waitlist |
| Gupta et al., 2021 – India | Healthcare workers caring for COVID–19 patients | Hospital-based | 35 (18:17) | Not reported but all participants were under 30 years old. | Brief eclectic psychotherapy | Treatment as usual with information on Covid prevention and control |
| Hosseinzadeh Asl , 2022 – Turkey | Social workers involved in COVID–19 response | Community-based clinics | 59 (30:29) | 33.1 | Mindfulness-based cognitive therapy (MBCT) | Waitlist |
| Khosravi et al., 2022 – Iran | Pregnant women affected by COVID–19 | In two community-based health centres | 66 (33:33) | 26.1 and 28.4 intervention and control group | Individual counselling | Treatment as usual |
| Li H, et al., 2023 – China | COVID–10 patients | Hospital-based | 58 (29:29) | 37.9% middle-aged and 43.1% young age. | Online Mindfulness-Based Stress Reduction (MBSR) | Conventional psychological counselling |
| Li et al., 2020 – China | COVID–19 Patients | Hospital-based | 94 (47:47) | 48 | Cognitive-behavioural therapy with cognitive intervention, relaxation techniques, problem-solving and social support strategy | Treatment as usual – received routine treatment and nursing care |
| Liu et al., 2021 – China | COVID–19 patients | Hospital-based | 140 (70:70) | 43.8 | Group psychological intervention and pulmonary rehabilitation exercises | Treatment as usual – received routine care as per COVID–19 protocols |
| Liu et al., 2021 – China | COVID–19 patients | Hospital-based | 273 (137:136) | Not reported | Computerized CBT (cCBT) included relaxation mental imagery training and mindfulness | Treatment as usual – received usual care per COVID–19 protocols |
| Mirhosseini et al., 2022 – Iran | COVID–19 survivors | Hospital-based | 70 (35:35) | Three-fourth were above 40 years old | Psychoeducational support training program | Treatment as usual – received routine care |
| Shabahang et al., 2021 – Iran | College students affected by COVID–19 | University-based | 152 (76:76) | 24.7 | A video-based cognitive–behavioural therapy | Waitlist |
| Shaygan et al., 2021 – Iran | COVID–19 patients | Hospital-based | 50 (27:23) | 36.8 | Online multimedia psychoeducational intervention | Telephone-based multimedia psychoeducational interventions |
| Shaygan et al., 2023 – Iran | COVID–19 patients | University Hospital | 72 (36:36) | Most participants were in the range of 30–50 years old | Psychoeducational intervention that involved coping techniques, positive thinking and relaxation | Treatment as usual – received routine care |

**Table 1.** (*Continued*)

| Author, year – country | Study population | Settings | N (intervention vs control) | Mean age (yrs) | Interventions | Controls |
|---|---|---|---|---|---|---|
| Sun et al., 2022 – China | College students in COVID–19 quarantine | In quarantine facilities | 114 (57:57) | 22.2 | Mindfulness-based intervention | Social Support-based mHealth |
| Zhou et al., 2022 – China | Nurses involved in COVID–19 response | Hospital-based | 118 (60:58) | 29.6 | E-aid cognitive behavioural therapy | Waitlist |

The effectiveness of these trials varied from non-significant change to high effect sizes in reducing depression, anxiety, stress, sleep problems and post-traumatic symptoms. The majority of the trials showed a significant reduction in depression, anxiety, stress and insomnia scores between baseline and post-treatment assessments. Brief psychoeducational interventions based on cognitive restructuring, mindfulness, relaxation and stress management techniques were effective in reducing perceived stress and anxiety symptoms, and to improve resilience and self-efficacy (Shaygan et al., 2023; Shaygan et al., 2021; Mirhosseini et al., 2022; Alkhawaldeh, 2023). Also, brief mindfulness-based interventions (Sun et al., 2022) and mindfulness-based stress reduction (Li et al., 2023) treatments were effective in addressing depression, anxiety and generalised anxiety disorder. Remotely delivered CBT and mindfulness-based CBT interventions showed promising but non-significant changes in reducing depression, anxiety, sleep and stress (Hossain et al., 2020; Ghazanfarpour et al., 2022; Li et al., 2020; Liu et al., 2021; Gupta et al., 2021). Although NET had a statistically significant change in reducing post-traumatic stress symptoms, there was non-significant change in sleep quality, depression and anxiety scores (Fan et al., 2021). Similarly, brief Eclectic Psychotherapy (Gupta et al., 2021) and Individual Counselling (Khosravi et al., 2022) were non-effective in bringing significant changes in anxiety depression and perceived stress (Table 2).

Anxiety Sensitivity Index (ASI); BMSF (burnout measure short-from); CD-RISC (Connor-Davidson Resilience Scale); CVAQ (COVID-19 Anxiety Questionnaire); DASS (Depression, Anxiety, Stress Scale); $DASS^D$ (DASS-depression); $DASS^A$ (DASS-anxiety); $DASS^S$ (DASS-stress); HADS (Hospital Anxiety and Depression Scale); GAD (Generalized Anxiety Disorder); $HADS^A$ (HADS-anxiety); $HADS^D$ (HADS-depression); HAMA (Hamilton Anxiety Rating Scale); HAMD (Hamilton Depression Rating Scale); PREPS (Pandemic-Related Pregnancy Stress Scale); PCL-C (PTSD Checklist Civilian version); PHQ (Patient Health Questionnaire); PSS (Perceived Stress Scale); PSQI (Pittsburgh Sleep Quality Index); PTSS (Post-Traumatic Stress Symptoms); SAS (State Anxiety Scale); SDS (Self-rating Depression Scale); SHAI (Short Health Anxiety Inventory); STAI (State-Trait Anxiety Inventory); SRAS (Self Rating Anxiety Scale); SMD (Standard Mean Difference); SUD (Subjective Unit of Distress); SUPPH (Strategy Used by People to Promote Health).

### *Quality of the included studies*

Of 17 included trials, 14 (82.3%) had at least one unclear domain with respect to the Cochrane Collaboration risk of bias checklist and 10 (n=10, 58.8%) had at least one domain with high risk of bias (Figure 2). Only one trial had a low risk of bias in all Cochrane risk of bias assessment items (Sun et al., 2022). In most trials, blinding of study participants and outcome assessors was not carried out. Similarly, several trials had a recruitment bias due to not employing proper randomisation (Figure 2).

### Discussion

This systematic review explored several electronic databases to identify and summarise RCTs that were conducted in LMICs, and to synthesise evidence on the effectiveness of psychological interventions in addressing mental health issues during infectious disease outbreaks. After a robust systematic search and careful screening, we found 17 RCTs eligible for this systematic review. These trials were all conducted during the COVID-19 pandemic from 2020 to 2023, showing the huge research gap before the COVID-19 pandemic in LMICs, despite many examples of devastating outbreaks. This systematic review found no included trials conducted in Africa or Latin America, again despite there being many examples here. Notably, most of the included trials were delivered remotely, despite there being very little robust evidence of this means of delivering treatments at the time. The trials were of interventions to address depression, anxiety, stress, sleep and post-traumatic stress symptoms among COVID-19 patients, frontline healthcare professionals involved in COVID-19 response, and college students in quarantine. A range of interventions were used including CBT, psychoeducational interventions, mindfulness techniques, NET and individual counselling with varying number of sessions and duration.

The systematic review found that brief psychoeducational interventions based on cognitive, relaxation and stress management techniques were effective for management of perceived stress and anxiety symptoms as well as in improving resilience, coping strategies and self-efficacy (Shaygan et al., 2023; Shaygan et al., 2021; Mirhosseini et al., 2022). These findings align with a report from a single-blind RCT in a high-income country (Morina et al., 2023) in which a brief psychoeducational intervention was successful in reducing psychological distress, generalised worry and burnout among healthcare workers during the COVID-19 pandemic in Zurich, Switzerland. This trial recommended booster sessions to maintain the initial gains beyond six months. Brief psychoeducational interventions are non-intensive, flexible and can be delivered by non-specialists in any context including in outbreaks/pandemics.

Moreover, brief mindfulness-based intervention (Sun et al., 2022) and mindfulness-based stress reduction interventions (Li et al., 2023) were effective in addressing depression, anxiety and generalised anxiety disorder. Similar findings have been reported from a systematic review and meta-analysis of 26 RCTs that mindfulness-based interventions reduced depressive

**Table 2.** Interventions, key findings and important limitations (N=17)

| Author, year – country | Outcomes | Outcome measures | Intervention, therapists, sessions | Assessment point, attrition rate | Key results and important limitations |
|---|---|---|---|---|---|
| Alkhawaldeh, 2023 – Jordan | Occupational stress Coping strategy | Nursing stress scale Brief COPE questionnaire | **Psychoeducational intervention.** Six sessions of psychoeducational intervention delivered over 2 weeks for 2 hours by a **trained psychiatrist face-to-face** in interactive learning approach. | Assessments were done at baseline, immediately after the last session and 1 month after the last session. **Retention**: 80/84 (95.3%); 40/42 in intervention and 40/42 in the control groups completed. | The degrees of occupational stress and coping strategies significantly differed between study groups over the three points of assessment. The **psychoeducational interventional programme was a valuable noninvasive** method that can improve individual coping strategies to manage stress in practice during the COVID–19 pandemic. **Limitation**: small sample size and no longer follow-up involved. |
| Dincer and Inangil, 2021 – Turkey | Psychological Distress Anxiety Burnout | SUD SAS BMSF | **Emotional Freedom Techniques (EFT)** intervention. A single session of emotional freedom training delivered 20 minutes with practical sessions – **online** in groups. Treatment was **delivered by trained personnel in EFT**. | Before-after assessment: pre-test and post-test assessment was done before and at the end of the session through SurveyMonkey among **frontline nurses**. **Retention**: 100% | **Statistically significant reductions in stress, anxiety and burnout** observed. A single online group EFT session reduced stress, anxiety and burnout levels in nurses treating COVID–19. Pre-test and post-test assessments were done within a short time interval, after a single session intervention. **Limitation**: No follow-up assessment was done to inform sustained response. |
| Fan et al., 2021 – China | PTS symptoms Depression Anxiety Sleep quality | PCL-C SDS SRAS PSQI | **Narrative exposure therapy (NET)**. The NET therapy had duration of eight weeks, with one or two sessions a week, lasting for 90–120 minutes each time involved 6-months follow-up. Sessions were delivered in a one-to-one model by **certified Doctors and Nurses** via the **internet**, mobile phones, WeChat and later in person. | Before-after assessment: **COVID–19 patients** were assessed before and after the last session of the intervention. **Retention**: 100% in both groups. | **Statistically significant change in PTSS** was found. There were non-significant improvements in sleep quality, anxiety and depression score. **Limitations**: Relatively small sample size, selection bias (only those with internet connection were included in the study), assessors were not blind and the PCL-C tool not widely used in China. |
| Ghazanfarpour et al., 2022 – Iran | Anxiety Depression | HADS[A] HADS[D] | **Cognitive-behavioural and mindfulness-based Techniques.** Seven sessions counselling was implemented through voice or video calls, text chats and video clips shared on WhatsApp, in seven sessions on seven consecutive days – delivered by trained **MSc students in midwifery counselling**. Each session lasts 45–90 minutes – **tele-counselling.** | Before-after assessment: pre-test and post-test assessment done **among healthcare workers**. **Retention**: 103/109 (94.5%); 50/53 in the intervention group and 53/56 control group lost follow-up due to workload and infection. | A promising result was observed in reducing anxiety and depression related to the Corona virus. Pre- and post-assessment in the intervention group showed **a significant reduction anxiety and depression**. However, change between the intervention and control group at the end of the intervention was non-significant. **Limitation**: Generalizability of the results is weak due to recruitment bias. |
| Gupta et al., 2021 – India | Depression Anxiety Stress | DASS–21 | **Brief Eclectic Psychotherapy using tele-counselling**. The intervention involved three sessions included expressing empathy, emphasizing on strengthening, psychoeducation on relaxation and motivational interviewing sessions delivered through telephonic audio conversation combined with WhatsApp and email messaging. Each session lasts for 30-minute over 7–10 days – **online telecounselling**. No information on the delivering agents. | Point of assessment was not clearly defined. Frontline health workers were assessed overtime until the completion of the intervention. **Retention**: 24/29 (82.8%); 11/14 intervention and 13/15 control arm. | A significant **over-time-effect was observed** depression, anxiety and stress. However, there was no significant between the two groups overtime. **Limitation**: there was a high refusal rate in the recruitment of participants into the study and a high attrition rate, so selection bias could not be ruled out. Sample size was not powered, the assessor was not blind and the tool was not validated. |
| Hosseinzadeh Asl, 2022 – Turkey | Depression Anxiety Stress Self-compassion | DASS–21 & self-compassion scale | **Mindfulness-based cognitive therapy (MBCT) and meditation.** Four weekly 70-min mindfulness training sessions plus 10 to 20 min of daily meditation as homework. Sessions were delivered **online via zoom app**. | Pre-test, post-test and follow-up assessment after 1 month were done **among frontline social workers**. **Retention**: 49/59 (83.1%); (28/30 in the experimental group and 21/29 in the control group) | Brief MBCT for 4 weeks improves psychological flexibility, self-compassion and depression in social workers, **but not effective in reducing anxiety and stress**. The effectiveness of the brief online MBCT sustained at least for one month after the interventions completed. **Limitation**: low generalizability of the results due to recruitment bias due to the use of convenience sampling. And mechanism of change was not examined. |

(Continued)

**Table 2.** (*Continued*)

| Author, year – country | Outcomes | Outcome measures | Intervention, therapists, sessions | Assessment point, attrition rate | Key results and important limitations |
|---|---|---|---|---|---|
| Khosravi et al., 2022 – Iran | Stress of self and the fetus | PREPS–15 | **BELIFE individual counseling** that shapes the current expectations of women and their feelings about pregnancy tensions. Individual counseling sessions provided as part of antenatal care in three 60-minute sessions, each with 1-week interval – phone calls were included between sessions – **hybrid (face-to-face and phone call)**. | Post-test assessment was done 2 weeks after the last counseling session among **COVID–19 affected pregnant women**. **Retention**: 100% attendance rate. | Although the **individual counselling was able to reduce the mean scores of stress** of Covid-19 in the experimental group, this difference was not statistically significant. **Limitation**: recruitment bias |
| Li et al., 2023 – China | Anxiety Depression | SRAS SDS | **Mindfulness-Based Stress Reduction (MBSR)**. The mindfulness practice was performed in 30 minutes per session, 2 sessions daily (before nap and nightfall) for 5 days – online using audio-video mindfulness designs. | Pre-test and post-test evaluation was done. Post-test was assessed at the end of the intervention among COVID–19 patients. **Retention**: not reported. | **Online-based MBSR intervention alleviated anxiety and depression symptoms** among COVID–19 patients in quarantine. Online MBSR found to be a cost-effective and time-efficient interventions. **Limitation**: long-term effects of online-based MBSR, allocation bias and matching of study subjects at baseline was not ensured and sample size was not powered to detect effectiveness. |
| Li et al., 2020 – China | Depression Anxiety Stress | DASS–21 | **Cognitive-behavioural therapy (CBT)** with cognitive intervention, relaxation techniques, problem-solving and social support strategy. CBT was delivered once a day for 30 minutes. Depending on the length of hospital stay (Average 14.4 days). **CBT trained Nurses** facilitated sessions – face to face. | Baseline and post-intervention assessment was done among COVID–19 patients. **Retention**: 47/47 in the intervention group (100% attendance) and 46/47 in the control group. | All participants in the intervention group **had a significant reduction in depression, anxiety and stress status, but there were no significant** differences between the intervention and control groups. CBT was effective in improving psychological health including depression, anxiety and stress among patients with COVID–19. **Limitation**: 1) relatively short period of intervention with no long-term follow up after the completion of the intervention therefore lead to misinterpretation of the effectiveness of the intervention; 2) small sample size due to shortage of therapists and rapid transmission of the infection. |
| Liu et al., 2021 – China | Anxiety Sleep quality | SAS PSQI | Group **psychoeducational intervention and pulmonary rehabilitation exercises**. Psychological interventions delivered using WeChat Groups and instructional videos – online | Assessments were carried out at baseline and post-intervention. **Retention**: not reported. | Both **anxiety and poor sleep quality scores of the intervention group were significantly lower than** those of the control group. This intervention was useful to mitigate anxiety and sleep disorders for the patients with mild COVID–19 infections |
| Liu et al., 2021 – China | Anxiety Depression | HAMA HAMD | Computerized **CBT (cCBT)** included Relaxation mental imagery training and Mindfulness meditation. Intervention was delivered through more than 10 minutes of self-directed individual therapy per day for 1 week – a self-help remote intervention model using iPad. | Pre- and postintervention assessments. Follow-up assessments were done again within 1 month after the post-intervention assessment. **Retention**: 252/273 (92.3%); 126/136 in the intervention and 126/137 in the control group completed. | Computerised **CBT program had a significant effect in relieving symptoms of anxiety, depression and insomnia** at post-intervention and follow-up assessment among patients with COVID–19. However, the insomnia symptoms in females and those with middle school education were not improved. **Limitation**: participants were non-blind for the intervention, the sample sizes were relatively small and the time before the follow-up was relatively short. |
| Mirhosseini et al., 2022 – Iran | Perceived stress | PSS–14 | **Psychoeducational support** training program. Six online psychoeducational group sessions were delivered on stress management. Each session last for 35–45 minutes once in a week – online group video calls via WhatsApp. | Pre- and post-intervention assessments were done. **Retention**: 100% attendance rate in both groups. | A statistically significant reduction in perceived stress score observed in the intervention group at post-intervention assessment. Using an online psychoeducational support group is suggested as a **useful and low-cost** solution to relieving the psychological stress of caregivers of COVID–19 survivors. **Limitation**: generalizability of the results to other contexts is limited. |

(*Continued*)

| Author, year – country | Outcomes | Outcome measures | Intervention, therapists, sessions | Assessment point, attrition rate | Key results and important limitations |
|---|---|---|---|---|---|
| Shabahang et al., 2021 – Iran | Anxiety | CVAQ SHAI ASI–3 | **Video-based CBT**. Intervention group received a CBT based self-help package of 9 video clips and 25-page online booklet. They were instructed to first watch a video clip for 15–20 minutes each and then read the corresponding 2–3 pages booklet for 3 days of each week over the course of 3 consecutive weeks – online multimedia. | Pre- and post-treatment evaluation among college students. **Retention**: 150/152 (98.7%); 75/76 in the intervention and similarly 75/76 in the control group | There was a **significant difference between the intervention and control groups** in COVID–19 anxiety, health anxiety, anxiety sensitivity and somatosensory amplification with small (0.2), medium (0.5) and large effect sizes (0.8) effect sizes respectively. Overall, the video-based CBT was slightly to moderately effective in lowering COVID–19 anxiety, health anxiety, anxiety sensitivity and somatosensory amplification of individuals with high levels of COVID–19 anxiety. **Limitation**: selection bias introduced due to convenient sampling, assessments were not masked, adherence to the intervention was not assessed and longer effect of the intervention was not assessed in follow-up. |
| Shaygan et al., 2021 – Iran | Resilience and Perceived stress | CD-RISC, Perceived Stress Scale | Online multimedia **psychoeducational intervention**. An online multimedia psychoeducational intervention delivered for 2 weeks. The interventions consisted of 14 daily modules and patients were asked to complete 1 module per day, which was designed to be 60 min in total. Each module consists of videos, audios and text files – online multimedia. | Pre- and post-treatment assessed before and 2 weeks after the interventions were done among **COVID–19 patients**. **Retention**: 48/50 (96%) 26/27 in the treatment and 22/23 in the control groups completed the post-treatment assessment. | Compared with the control groups, patients in the online multimedia **psychoeducational intervention had a greater score of resilience and reduced level of stress** after 2 weeks. The online multimedia psychoeducational intervention based on CBT techniques, mindfulness-based stress reduction and positive psychotherapy has shown significant benefits and can be regarded as a **cost-effective and convenient** tool to protect the patients from the stress. **Limitation:** small sample size, lack of long-term follow-ups and was no objective measure of adherence. |
| Shaygan et al., 2023 – Iran | Self-efficacy Anxiety | SUPPH–29 STAI | **Psychoeducational intervention** delivered via WhatsApp groups daily for 14 days until the quarantine period is over. Video, audio and text files were shared on WhatsApp. Psychologists, mental health nurses and psychiatrists involved in the delivery of the sessions – online multimedia. | Pre- and post-treatment assessed before and 2 weeks after the intervention among **COVID–19 patient**s. **Retention**: 100% attendance in both groups. | The **intervention was effective in reducing self-efficacy and anxiety**. Interactive psychoeducational interventions via social networks are **cost-effective treatments** that can improve self-efficacy and educed anxiety among patients infected with COVID–19 who lived in home quarantine. **Limitation**: limited generalisability of the results. |
| Sun et al., 2022 – China | GAD Depression | GAD–7 PHQ–9 | **Mindfulness-based intervention**. 60-minutes sessions per week for 4 weeks. App-based delivery using instructional Video – mobile app based. | Baseline, immediate post-intervention (1 month) and at follow-up (2-month post-baseline) assessments were done. **Retention:** >80% attendance in both groups. | Compared with social support mental health intervention, **mindfulness-based intervention had superior effect on anxiety and both conditions improved depression**. Mindfulness intervention demonstrated to be cost-effective, more feasible and acceptable in program engagement, evaluation, skills improvement, and perceived benefit and to address anxiety and depression. **Limitation**: the results may not guarantee effectiveness in the real world. |
| Zhou et al., 2022 – China | Sleep quality GAD Depression | PSQI GAD–7 PHQ–9 | **E-aid CBT**. CBT courses involving relaxation training communicated with healthcare providers online via mobile phone or tablets for 6 weeks – online. | Pre- and post-treatment evaluation was done among frontline nurses, after 6 weeks of intervention. **Retention**: 100% in both groups | Compared with the scores of the control group, **sleep quality improved significantly** among the participants in the treatment group. The GAD–7 and PHQ–9 scores in the eCBT-I group were significantly lower after treatment than before treatment. Compared with subjects in the control group subjects in the eCBT-I group had lower scores on the GAD–7 and PHQ–9 scales after treatment. E-CBT improved the sleep quality of frontline nurses during the COVID–19 prevention and control period and relieved anxiety and depression. **Limitation**: most study participants were women, so the results are not fully generalised. |

Key: N*, number of participants completed the study.

| Author (year) | Selection Bias | | Reporting Bias | Performance Bias | Detection Bias | Attrition Bias |
|---|---|---|---|---|---|---|
| | Random Sequence Generation | Allocation Concealment | Selective Reporting | Blinding (participant and personnel) | Blinding (outcome) | Incomplete Outcome Data |
| Alkhawaldeh JM, 2023 – Jordan | Low | Low | Low | High | High | Low |
| Dincer B, and Inangil D, 2021 – Turkey | Low | High | Unclear | High | Low | Low |
| Fan Y, et al., 2021 – China | Unclear | High | Low | Unclear | High | Low |
| Ghazanfarpour M, et al., 2022 – Iran | Low | Unclear | Low | Unclear | Unclear | Low |
| Gupta S, et al., 2021 – India | Low | High | Unclear | Low | High | High |
| Hosseinzadeh Asl NR., 2022 – Turkey | Unclear | Unclear | Low | High | High | Low |
| Khosravi HM, et al., 2022 – Iran | Unclear | Unclear | Low | Unclear | Unclear | Low |
| Li H, et al., 2023 – China | Low | Low | Low | Low | Unclear | Unclear |
| Li J, et al., 2020 – China | Low | High | Low | High | Low | Low |
| Liu Y, et al., 2021 – China | Low | High | Unclear | Low | High | Low |
| Liu Z, et al., 2021 – China | low | Low | Low | High | High | Unclear |
| Mirhosseini S, et al., 2022 – Iran | High | High | Low | Unclear | Low | Low |
| Shabahang R, et al., 2021 – Iran | Low | Unclear | Low | Low | Low | Low |
| Shaygan M, et al., 2021 – Iran | Low | Low | Unclear | Low | Unclear | Low |
| Shaygan M, et al., 2023 – Iran | High | Unclear | Low | Unclear | Low | Unclear |
| Sun F, et al., 2021 – China | Low | Low | Low | Low | Low | Low |

**Figure 2.** Risk of bias assessment for included trials using the Cochrane Collaboration's Risk of Bias tool (N=17).

symptoms significantly among adults affected by COVID-19 pandemic (Fu et al., 2024).

Remotely delivered CBT and mindfulness-based CBT interventions showed promising but non-significant changes in reducing depression, anxiety, sleep and stress (Hossain et al., 2020; Ghazanfarpour et al., 2022; Li et al., 2020; Liu et al., 2021; Gupta et al., 2021). Although CBT has superior benefits and is a first-line treatment for a variety of mental health conditions (Surmai and Duff, 2022), it may be more effective when it is provided intensively for longer sessions (over 12 sessions) over longer period of time (Levy et al., 2020). Evidence shows that people who are taking CBT have shown a more gradual curse of change (Driessen and Hollon, 2010), and the minimum number of sessions needed to address common mental health problems is between 7 and 14 sessions (Robinson et al., 2020). Additionally, the use of active treatments (e.g., in standard interventions) for controls could also result in non-significant changes for CBT (Cuijpers, 2024).

Although, NET was superior in reducing post-traumatic stress symptoms to the control group, there was non-significant change in improving sleep quality, reducing depressive and anxiety symptoms (Fan et al., 2021). NET is one of the recommended therapies for the prevention and treatment of post-traumatic stress disorder (Megnin-Viggars et al., 2019), and the results of our systematic review showed that NET is effective in reducing post-traumatic stress symptoms among COVID-19 patients. Even so, the therapeutic components of NET are designed to resolve traumatic symptoms, its broader efficacy beyond PTSD requires further investigation.

Due to their simplicity and adaptability, these interventions have been recommended as appropriate to be delivered in global normative guidelines for some time, but their adaptation for delivery using different approaches including hybrid face-to-face and online, or via phone or video calls was novel and often brought about by necessity rather than being well established in evidence. The use of digital platforms in most included trials to deliver interventions remotely was deemed appropriate in outbreak/pandemic contexts, given contact limitations and scale of demand. This seemed to have proven to be acceptable, as evidenced by the high recruitment and completion rate, where more than 90% of participants completed all sessions in 80% (n=12) of the included trials. Despite the lack of access to digital technologies and low digital literacy in low-income settings, delivering interventions remotely using flexible approaches and multimedia platforms could strengthen the uptake of interventions as well as promote infection prevention and control in outbreaks/pandemics.

The current systematic review has several implications in filling the evidence gap in understanding how to effectively address mental health needs during infectious disease outbreaks. The lack of inclusion of issues related to culture in these studies on evidence-based practice is concerning. This is a topic that is often identified as important, and in fact efforts at adaptation, or even locally developed practice embedded in local cultures, are common, so there is a need for a high-quality research for informed decisions to equip health systems with more treatment options that properly incorporate sensitivity to culture during disease outbreaks. Importantly, it found some evidence for the value of established psychological interventions in what was a unique set of circumstances, requiring innovative approaches to delivery in LMICs. It found that these were often feasible and acceptable, with high adherence, though there may be bias associated with being part of a study. However, there are several limitations that need to be considered when interpreting the results of this systematic review. These are, but

not limited to: (i) as the result of lack of consistency, for example in standard case definition and outcome measures, and lack of clarity on the cultural validation of the tools, it may be difficult to generalise effectiveness of results to other populations and contexts; (ii) most of the included trials did not examine sustained effectiveness and therefore longer-term effectiveness of the interventions is unknown; (iii) trials that were written and published in languages other than English were not included in this review and (iv) the overall quality of evidence from these trials is moderately high, although the quality of evidence from each trial varies significantly. Weaknesses of included trials included a lack of proper randomisation, blinding and small sample sizes.

## Conclusion

Non-specialist delivered brief psychological interventions is likely to be valuable for addressing the huge mental health needs that arise in outbreaks. Overall, this review demonstrated that brief and remotely delivered psychoeducational interventions seem effective, feasible, cost-effective and time-efficient in the context of the COVID-19 pandemic, which provides valuable insights into their use in future outbreaks. The huge evidence gap in LMICs was marked – none of the included trials were from Africa and Latin America – despite Africa being where outbreaks are most common. Hence, addressing the huge research gap should be a priority to inform evidence-based and resource-efficient psychological interventions for outbreak/pandemic contexts in LMICs. While it was appropriate to innovate rapidly during the exceptional circumstances of the COVID-19 pandemic, future research should examine the use, applicability and scalability of digital interventions in LMICs, to better inform future outbreak preparedness and response. A particular consideration should also be given to the cultural adaptation of psychological interventions and mental health tools, in the context of still centralised production of normative guidance, which draws largely on evidence from high-income countries.

**Open peer review.** To view the open peer review materials for this article, please visit http://doi.org/10.1017/gmh.2025.22.

**Supplementary material.** The supplementary material for this article can be found at http://doi.org/10.1017/gmh.2025.22.

**Data availability statement.** All data relevant to the study are included in the article or uploaded as supplementary information.

**Acknowledgements.** BAY, JE and MA are paid salaries from the UK Public Health Rapid Support Team (UK-PHRST) and they would like to thank the UK-PHRST for the time and resources devoted to this research. The UK-PHRST is funded by UK Aid from the Department of Health and Social Care and is jointly run by the UK Health Security Agency and the London School of Hygiene & Tropical Medicine. The views expressed in this publication are those of the authors and not necessarily those of the Department of Health and Social Care or other institutions with which the authors are affiliated.

**Author contributions.** BAY, JE and JAS conceptualised the study. BAY and JAS conducted literature search, article screening and selection. BAY and JAS did quality appraisal of the included studies. JE and MA assisted the data collection and synthesis. BAY drafted the original writing and JE, JAS and MA reviewed and edited it subsequently.

**Competing interest.** The authors declare that they have no conflict of interest.

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
