## [Reviewer Report]

This is a useful review, and generally well written.

The term psychosocial is used once (page 6) while psychologological is used throughout, and this needs explanation. A reason for excluding psychosocial interventions (e.g. social support) and only psychological would be helpful.

The title refers to LMICs, but none of the included settings are low income, so I suggest the title should only include middle income countries, and this should also be made clear in the article.

There is a lack of information and discussion about the control groups, e.g. whether non significant findings could be due to equally good treatment in the control condition. Both the abstract and impact statement mention culturally appropropratie interventions, but there is little mention of this is the actual review, thus the empahsis on culture needs justification. More discussion of the impact of the findings for Latin America and Africa, and especially for low income countries, would be useful.

Page 6 line 106 should probably be workers, not works.

the sentence page 7 line 147 should probably read published other languages than in English.

---

## [Editor Report]

Thank you for your submission. On review this is well executed, with valuable findings for the global mental health community. Kindly attend to/rebut minor comments of second reviewer. Please remove key to measures under Table 1 as it only pertains to Table 2.